# Phylogenetic Analysis with Prediction of Cofactor or Ligand Binding for *Pseudomonas aeruginosa* PAS and Cache Domains

Andrew Hutchin,[a,b,c,d*] Charlotte Cordery,[a,b,c,e] Martin A. Walsh,[b,c] Jeremy S. Webb,[a,e] Ivo Tews[a,e]

[a]Biological Sciences, Institute for Life Sciences, University of Southampton, Southampton, United Kingdom
[b]Diamond Light Source, Harwell Science and Innovation Campus, Didcot, United Kingdom
[c]Research Complex at Harwell, Harwell Science and Innovation Campus, Didcot, United Kingdom
[d]Structure and Function of Biological Membranes Lab, Université Libre de Bruxelles, Brussels, Belgium
[e]National Biofilms Innovation Centre, University of Southampton, Southampton, United Kingdom

**ABSTRACT** PAS domains are omnipresent building blocks of multidomain proteins in all domains of life. Bacteria possess a variety of PAS domains in intracellular proteins and the related Cache domains in periplasmic or extracellular proteins. PAS and Cache domains are predominant in sensory systems, often carry cofactors or bind ligands, and serve as dimerization domains in protein association. To aid our understanding of the wide distribution of these domains, we analyzed the proteome of the opportunistic human pathogen *Pseudomonas aeruginosa* PAO1 *in silico*. The ability of this bacterium to survive under different environmental conditions, to switch between planktonic and sessile/biofilm lifestyle, or to evade stresses, notably involves c-di-GMP regulatory proteins or depends on sensory pathways involving multidomain proteins that possess PAS or Cache domains. Maximum likelihood phylogeny was used to group PAS and Cache domains on the basis of amino acid sequence. Conservation of cofactor- or ligand-coordinating amino acids aided by structure-based comparison was used to inform function. The resulting classification presented here includes PAS domains that are candidate binders of carboxylic acids, amino acids, fatty acids, flavin adenine dinucleotide (FAD), 4-hydroxycinnamic acid, and heme. These predictions are put in context to previously described phenotypic data, often generated from deletion mutants. The analysis predicts novel functions for sensory proteins and sheds light on functional diversification in a large set of proteins with similar architecture.

**IMPORTANCE** To adjust to a variety of life conditions, bacteria typically use multidomain proteins, where the modular structure allows functional differentiation. Proteins responding to environmental cues and regulating physiological responses are found in chemotaxis pathways that respond to a wide range of stimuli to affect movement. Environmental cues also regulate intracellular levels of cyclic-di-GMP, a universal bacterial secondary messenger that is a key determinant of bacterial lifestyle and virulence. We study *Pseudomonas aeruginosa*, an organism known to colonize a broad range of environments that can switch lifestyle between the sessile biofilm and the planktonic swimming form. We have investigated the PAS and Cache domains, of which we identified 101 in 70 *Pseudomonas aeruginosa* PAO1 proteins, and have grouped these by phylogeny with domains of known structure. The resulting data set integrates sequence analysis and structure prediction to infer ligand or cofactor binding. With this data set, functional predictions for PAS and Cache domain-containing proteins are made.

**KEYWORDS** Cache domain, PAS domain, phylogeny, *Pseudomonas*, cofactors, phylogenetic analysis, sensory transduction processes

Address correspondence to Ivo Tews, ivo.tews@soton.ac.uk.

*Present address: Andrew Hutchin, Evotec (UK) Ltd., Abingdon, United Kingdom.

The authors declare no conflict of interest.

The Gram-negative bacterium *Pseudomonas aeruginosa* is capable of growth in a wide range of different conditions, including soil and coastal marine habitats or plant and animal tissues (1, 2). *P. aeruginosa* is also a significant opportunistic human pathogen recently described as a species urgently requiring development of novel antibiotics for treatment of disease due to the emergence of multidrug-resistant strains (3). *P. aeruginosa* is able to infect patients suffering from burns, immunosuppression, and cystic fibrosis (CF); reduced pulmonary function caused by chronic *P. aeruginosa* infection is the largest cause of mortality in cystic fibrosis patients (1, 2, 4).

Diversity in cultivation of habitats is likely underpinned by adaptation mechanisms of *P. aeruginosa* to alter phenotypic behavior. This marked pleiotropism identifies a broad array of environmental cues and a number of archetypal bacterial responses. These resulting bacterial responses might include movement away from or toward a specific chemical stimulus, also known as chemotaxis (5). Alteration of gene expression may also be directly induced by a stimulus, often as part of a two-component regulatory system (6). Finally, transition from a planktonic phenotype to a sessile biofilm lifestyle makes *P. aeruginosa* an important human pathogen causing chronic infection (7, 8). This transition, and with it, bacterial virulence, is critically regulated by intracellular c-di-GMP levels (9). *P. aeruginosa*, and particularly the reference strain PAO1, is an extensively studied model organism in biofilm formation (2, 10, 11).

Adaptive responses that require mechanisms of signal perception are often directly transmitted through sensory proteins. A classic, versatile, and very widespread protein architecture used in many sensory proteins is the Per-Arnt-Sim or PAS domain (12, 13). The domain was first identified to be conserved between the fly clock protein PERIOD, the vertebrate aryl hydrocarbon nuclear translocator (ARNT), and the fly developmental regulator single-minded (SIM) (14, 15), and PAS domains are found across all biological kingdoms. They are frequently found among bacterial sensory systems (15) and play crucial roles in environmental responses of *P. aeruginosa* (16, 17). They are also widespread in regulators of intracellular c-di-GMP, where they are suggested to play a role in regulation of virulence (18), as well as motility and biofilm phenotype (19).

Cache domains are the extracellular relatives of the intracellular PAS domains (20–22). Typically acting as signal receptors, they bind small ligands and propagate signals into the cell interior, suggested to be mediated by the C-terminal helix that crosses the membrane (21–24). They are often classified into sCache or dCache domains with one or two PAS-like domains, respectively (22). While Cache domains are the predominant superfamily of extracellular receptors in prokaryotes, they are also found as extracellular domains ubiquitous across all kingdoms (22).

Adaptation of PAS and Cache domains to a variety of signals is achieved by cofactor or ligand binding. Further, they can typically dimerize in response to physiological change and, in doing so, alter the activity of the effector outputs within PAS domain-containing proteins (14, 25). Many proteins contain several PAS domains that may be common in structure but different in function, and hence study of cofactor or ligand binding is a systems approach that is essential to determine how responses are regulated.

Here, we update the list of known PAS and Cache domains in *P. aeruginosa* PAO1, provide a phylogenetic analysis, and add functional insights using a sequence/structure-based approach, with prediction of cofactor or ligand binding. Building upon earlier identification through Hidden Markov Model analysis (22, 26), the analysis includes a total of 90 PAS domains, 2 sCache domains, and 9 dCache domains that were identified from a total of 70 genes. Phylogeny and physiology are put in context, as illustrated here for carboxylic acid-binding PAS domains.

## RESULTS

**Selection of *P. aeruginosa* PAO1 PAS and Cache domains.** The PAS domain fold is a highly conserved yet versatile structure. Sequence identity of PAS domains is typically below 20 percent (14, 15, 27), making the identification of all PAS domains in an organism difficult. Hidden Markov Models (HMM) were used as a sensitive method for

homology detection, and these methods are typically employed for cases with low sequence identity (28). Application of HMM methods through HMM-to-HMM comparison (implemented through HHblits [29]) has initially identified 106 domains from 70 PAO1 proteins (26), of which 18 form 9 dCache domains and 2 were later classified as sCache, thus leaving 86 bona fide PAS domains (22).

The search used here employed domain boundaries predicted from the earlier studies. Additionally, the 70 PAO1 proteins were queried with the SMART domain prediction server (30, 31). The SMART analysis uses different criteria for minimum sequence length compared with the more stringent HMM-to-HMM analysis (26). SMART identifies the shorter PAS and PAC sequence motifs separately (32) and was thus able to identify the N-terminal PAS motif for PA4021, PA4112, PA4147, and PA4197. Sequences were extended at the C terminus to a length of ~120 amino acids to facilitate further analysis. The final list of 101 PAS and Cache domains selected from this analysis is given in Table 1.

**Phylogenetic analysis and grouping of PAS and Cache domains.** We performed a grouping of sequences by maximum likelihood phylogeny to understand the relationships between sequences (see Materials and Methods). Neighborhood within this phylogenetic analysis might infer similar properties with regard to cofactor or ligand binding. The relationship between proteins grouped within the same clade can thus be used as an indicator toward a functional assignment of the individual domains, leading to experimentally testable hypotheses.

The sequence alignment was performed against a reference data set made up from PAS and Cache domains with known ligand or cofactor obtained from structural analysis. Table 2 indicates cofactor or ligand identified in these structures and size of the ligand- or cofactor-binding pocket. The reference data set also included 37 sequences for ligand- or cofactor-free PAS or Cache structures, selected based on physiological roles not requiring ligand or cofactor, e.g., mediating dimerization or downstream signaling in response to a conformational change in a multidomain protein or as a result of binding to another protein. This ties in with the observation that a significant number of PAS or Cache domains were reported as structures without an associated cofactor or ligand (15).

The maximum likelihood phylogeny analysis with a 100-replicate bootstrap consensus tree is shown in Fig. 1. For the PAS and Cache domains analyzed here, we found that maximum likelihood grouped PAS or Cache domains from the reference data set into clades of similar cofactor or ligand binding across the largest number of bootstrap replicates, in comparison to other ways of constructing phylogenetic trees (see supplemental material for further detail). The phylogenetic tree identifies a number of clades and groups PAO1 sequences together with structurally known PAS and Cache domains. The grouping is based solely on the phylogenetic analysis and is thus unbiased by ligand or cofactor binding or structural knowledge. A number of PAO1 PAS and Cache domains have been characterized previously with respect to ligand or cofactor binding, and the fact that these sequences cluster in the tree with the reference sequence from structures with similar ligand or cofactor validates the approach taken here.

Assignments were made based on the basis of clustering in more than 15 bootstrap replicates, as this threshold provides unambiguous clustering of the reference PAS and Cache domains in almost all cases while retaining the clustering of known homologues. For previously uncharacterized PAO1 PAS or Cache domains, inference suggests that grouping of *P. aeruginosa* sequences with structural representatives from the reference indicates similar ligand or cofactor binding. Alignments of individual clades are presented in the supplemental material, and we give a few examples in the following section.

**Inferences from example clades and grouping of PAS and Cache domains.** A prominent clade, marked with arrow 1 in Fig. 1 (alignment found in the supplemental material), places the PAO1 PA1336, PA5165, and PA5512 dCache domains with the reference structure sequences of the two DctB dCache domains of *Vibrio cholerae* and

**TABLE 1** *P. aeruginosa* PAO1 proteins with PAS or Cache domains[a]

| Gene | Protein | Domain boundary | | | |
|------|---------|------|------|------|------|
| | | PAS1 | PAS2 | PAS3 | PAS4 |
| PA0172 | SiaA | dCache 102–304 | dCache 102–304 | | |
| PA0176 | Aer2/TlpG/McpB | 166–287 | | | |
| PA0285 | | 79–198 | 206–320 | | |
| PA0290 | | 31–151 | | | |
| PA0338 | | 50–170 | | | |
| PA0464 | CreC | sCache35–179 | | | |
| PA0533 | | 12–135 | 137–255 | 265–379 | |
| PA0575 | | 310–426 | 438–550 | 562–675 | 682–797 |
| PA0600 | AgtS | 323–436 | 446–568 | | |
| PA0847 | | 142–284 | 444–560 | | |
| PA0861 | RbdA | 243–363 | | | |
| PA0873 | PhhR | 82–187 | | | |
| PA0928 | GacS | 43–161 | | | |
| PA1098 | FleS | 74–164 | | | |
| PA1120 | TpbB/ YfiN | 46–152 | | | |
| PA1180 | PhoQ | 33–161 | | | |
| PA1181 | YegE | 298–415 | 427–542 | 553–674 | |
| PA1196 | DdaR | 20–132 | | | |
| PA1243 | | 57–169 | 343–456 | | |
| PA1261 | IhpR | 1–107 | | | |
| PA1336 | AauS | dCache51–346 | dCache51–346 | | |
| PA1347 | | 23–129 | | | |
| PA1423 | BdlA | 3–112 | 116–234 | | |
| PA1438 | MmnS | 41–166 | | | |
| PA1561 | Aer/ TlpC | 8–121 | | | |
| PA1611 | | 38–169 | | | |
| PA1930 | McpS | 17–134 | 139–254 | | |
| PA1976 | ErcS′ | 97–207 | 226–338 | 339–454 | |
| PA1992 | ErcS | 41–157 | | | |
| PA2005 | HbcR | 17–123 | | | |
| PA2072 | | 301–414 | | | |
| PA2177 | | 62–180 | 190–308 | | |
| PA2449 | | 79–182 | | | |
| PA2480 | | 30–148 | | | |
| PA2524 | CzcS | 34–171 | | | |
| PA2652 | CtpM | sCache42–198 | | | |
| PA2654 | TlpQ | dCache50–346 | dCache50–346 | | |
| PA2824 | SagS | 56–169 | | | |
| PA2870 | | 97–211 | 241–348 | | |
| PA3044 | RocS2 | 110–225 | | | |
| PA3271 | | 636–751 | | | |
| PA3946 | RocS1 | 573–687 | | | |
| PA4021 | EatR | 80–185 | 225–344 | | |
| PA4036 | | 432–537 | | | |
| PA4112 | | 343–460 | 491–614 | 626–744 | |
| PA4117 | BphP | 23–123 | | | |
| PA4147 | AcoR | 82–191 | 225–344 | | |
| PA4197 | BfiS | 158–265 | 266–383 | 389–504 | |
| PA4290 | | 411–520 | | | |
| PA4293 | PprA | 303–421 | 431–549 | 560–675 | |
| PA4307 | PctC | dCache34–275 | dCache34–275 | | |
| PA4309 | PctA | dCache35–273 | dCache35–273 | | |
| PA4310 | PctB | dCache35–274 | dCache35–274 | | |
| PA4398 | | 50–154 | 286–395 | | |
| PA4546 | PilS | 195–296 | | | |
| PA4581 | RtcR | 52–165 | | | |
| PA4601 | MorA | 290–411 | 582–705 | 717–845 | 825–967 |
| PA4633 | | dCache51–346 | dCache51–346 | | |
| PA4725 | CbrA | 630–739 | | | |
| PA4886 | | 69–166 | | | |
| PA4959 | FimX | 142–254 | | | |

**TABLE 1** (Continued)

| Gene | Protein | Domain boundary | | | |
|------|---------|------|------|------|------|
| | | PAS1 | PAS2 | PAS3 | PAS4 |
| PA4961 | | 53–166 | | | |
| PA4982 | AruS | 288–388 | | | |
| PA5017 | DipA | 9–130 | 344–460 | | |
| PA5124 | NtrB | 3–116 | | | |
| PA5165 | DctB | dCache44–291 | dCache44–291 | | |
| PA5361 | PhoR | 101–201 | | | |
| PA5442 | | 275–393 | 401–515 | | |
| PA5484 | KinB | 257–369 | | | |
| PA5512 | MifS | dCache31–298 | dCache31–298 | | |

[a]Of the 70 genes listed, several encode more than one PAS domain. Domain boundaries were identified by HMM analysis in previous studies (22, 26) or with the SMART domain web server (30, 31).

*Sinorhizobium meliloti* in 29 out of 100 bootstrap replicates. While PA5165 has previously been identified as a DctB homologue within *P. aeruginosa* (33, 34), this clade gives new insight, as it implies coevolution with PA1336 and PA5512. It may further predict the potential for binding similar ligands in all five domains.

A close relationship is detected between PA5124 PAS1 and the PAS domain from *Escherichia coli* DhaR, identified in 93 out of 100 bootstrap replicates, marked with arrow 2 in Fig. 1. The *E. coli* DhaR protein is a regulator of transcription. The PAS domain of DhaR contains a very small cavity that precludes binding of larger organic cofactors (Table 2), and the PAS domain is instead thought to be involved in signal transmission through dimerization (35). Conformational changes of the entire protein would be induced by binding of a number of different known interaction partners (35). By inference, the PA5124 PAS1 domain may also not bind any cofactor. This finding may be surprising, as the two proteins have vastly different domain architectures: DhaR consists of a GAF domain, a PAS domain, and a C-terminal domain involved in interaction with $\sigma^{70}$ (35), while PA5124 is predicted to have a single PAS domain, as well as a histidine kinase and an accompanying phospho-transfer domain known from two-component signaling pathways (6, 30, 31).

The phylogenetic tree in Fig. 1 shows many PAO1 PAS and Cache domains that do not cluster to the chosen reference data set, guided by choice of our reference cofactor- and ligand-binding domain data set. However, the phylogenetic analysis performed here still provides insight into the evolutionary origin of several of these domains. An example cluster identified in 96% of replicates is marked with arrow 3 in Fig. 1 and contains the two PAS domains of PA1423 (BdlA) and the two PAS domains of PA1930 (McpS). The protein architectures are similar, as both proteins consist of two PAS domains N-terminal to a methyl-accepting chemotaxis domain. PA1423 and PA1930 are unique within PAO1, as they possess methyl-accepting chemotaxis domains shorter than those of any other chemoreceptors (36). The similar architecture suggests functional differentiation for these proteins.

**Assignment of cofactor or ligand binding based on sequence motif.** The ability of a PAS or Cache domain to bind cofactor or ligand should be reflected in conservation of cofactor- or ligand-interacting amino acids. PAS and Cache domains are structurally homologous, albeit with overall rather low sequence identity. To add additional information, the alignment can therefore be constrained by predicted secondary structure. Using the combined primary and predicted secondary information then gives sufficient confidence for modeling of the 3D localization of conserved cofactor- or ligand-coordinating amino acids.

We use this approach here to inspect the ligand- or cofactor-binding environment. The PAO1 test data set was aligned to the different cofactor- and ligand-binding subsets of the reference data set, using secondary structure constraints through use of PROMALS3D (37). Detection of conservation or conservative substitution of amino acids known to form sidechain-mediated interactions within the resulting alignments

**TABLE 2** The reference data set contains sequences from PAS or Cache domain structures, grouped by physiological cofactor or ligand and by protein and species name, as well as references to the structural database and literature[a]

| Cofactor or ligand | $M_w$ of cofactor or ligand (g/mol) | Protein | Organism | PDB | PAS domain boundary from PDB RCSB | Pocket MS vol (Å³) | Comment |
|---|---|---|---|---|---|---|---|
| 4'-Hydroxycinnamic acid | 164.16 | Ppr | Rhodospirillum centenum | 1MZU (70) | 25–129 | 397.0 | |
| | 164.16 | PYP | Halorhodospira halophila | 2PHY (58) | 1–125 | 226.5 | |
| Autoinducers | 124.14 | VqmA | Vibrio cholerae | 6IDE (39) | 16–121 | 318.0 | |
| Aromatics | 92.14 | TodS | Pseudomonas putida | 5HWV (71) | 5–133 | 216.8 | |
| FAD | 785.55 | MmoS (PAS A) | Methylococcus capsulatus | 3EWK (72) | 1–100 | 763.2 | |
| | 785.55 | NifL | Azotobacter vinelandii | 2GJ3 (57) | 16–117 | 548.1 | |
| | 785.55 | Vivid | Neurospora crassa | 2PDR (73) | 35–149 | 691.8 | |
| Fatty acids | 228.37 | Caur_2278/ MltR | Chloroflexus aurantiacus | 3PXP (74) | 111–292 | 965.3 | |
| | 356.54 | HIF3a9 PAS-B | Homo sapiens | 4WN5 (75) | 235–343 | 1,108.0 | |
| | 200.32 | RpfR | Cronobacter turicensis | 6DGG (76) | 7–110 | 453.7 | |
| | 256.42 | Rv1364c | Mycobacterium tuberculosis | 3K3C (38) | 27–132 | 615.0 | |
| FMN | 456.34 | Aureochrome 1a LOV | Phaeodactylum tricornutum | 5A8B (77) | 34–138 | 552.8 | |
| | 456.34 | Cagg_3753 | Chloroflexus aggregans | 6RHG (78) | 48–152 | 612.1 | |
| | 456.34 | LOV | Dinoroseobacter shibae | 6GAY (79) | 32–141 | 858.9 | Pocket open to solvent |
| | 456.34 | EI222 | Erythrobacter litoralis | 3P7N (80) | 34–141 | 585.6 | |
| | 376.36 | EL346 (HTCC2694) | Erythrobacter litoralis | 4R38 (81) | 15–123 | 625.9 | Riboflavin binding |
| | 456.34 | Env1 | Hypocrea jecorina | 4WUJ (82) | 37–146 | 589.6 | |
| | 456.34 | LOV | Rhodobacter Sphaeroides | 4HIA (83) | 18–123 | 627.8 | |
| | 456.34 | LOV-HK | Brucella abortus | 3T50 (56) | 26–140 | 573.7 | |
| | 456.34 | NPH1-1 (LOV2) | Avena sativa | 2V0U (84) | 13–119 | 587.7 | |
| | 456.34 | AUREO1 | Ochromonas danica | 6I20 (85) | 16–120 | 633.0 | |
| | 456.34 | PAL PAS B | Nakamurella multipartita | 6HMJ (86) | 209–347 | 669.9 | |
| | 456.34 | Phot | Chlamydomonas reinhardtii | 1N9L (87) | 17–125 | 730.3 | |
| | 456.34 | Phot1 | Arabidopsis thaliana | 2Z6C (88) | 15–125 | 629.9 | |
| | 456.34 | Phot2 | Arabidopsis thaliana | 2Z6D (88) | 16–121 | 741.6 | |
| | 456.34 | Phy3 | Adiantum capillus-veneris | 1G28 (89) | 929–1032 | 715.6 | |
| | 456.34 | SB1-LOV | Pseudomonas putida | 3SW1 (90) | 16–119 | 1,075.2 | Pocket open to solvent |
| | 456.34 | AUREO1 | Vaucheria frigida | 3ULF (91) | 51–154 | 581.8 | |
| | 456.34 | YtvA | Bacillus subtilis | 2PR5 (25) | 8–111 | 738.5 | |
| | 456.34 | Ado1 LOV | Arabidopsis thaliana | 5SVG (92) | 16–129 | 602.0 | |
| Heme-B | 616.49 | Aer2 | Pseudomonas aeruginosa | 3VOL (93) | 32–135 | 814.2 | |
| | 616.49 | Aer2 | Vibrio cholerae | 6CEQ (94) | 170–280 | 1,007.5 | |
| | 616.49 | DosP | Escherichia coli | 1V9Y (95) | 30–132 | 565.1 | |
| | 616.49 | HODM | Pseudomonas mendocina | 5LTE (96) | 155–290 | 1,869.9 | |
| | 616.49 | FixL | Bradyrhizobium japonicum | 1DRM (97) | 13–117 | 984.8 | |
| | 616.49 | FixL | Rhizobium meliloti | 1D06 (98) | 26–130 | 907.4 | |

**TABLE 2** (Continued)

| Cofactor or ligand | Protein | $M_w$ of cofactor or ligand (g/mol) | Organism | PDB | PAS domain boundary from PDB RCSB | Pocket MS vol (Å³) | Comment |
|---|---|---|---|---|---|---|---|
| Heme-C | GSU0582 | 616.49 | *Geobacter sulfurreducens* | 3B47 (99) | 45–131 | 24.8 | Non-classical heme cofactor binding |
| | GSU0935 | 616.49 | *Geobacter sulfurreducens* | 3B42 (99) | 45–127 | 12.1 | |
| | TlI0287 | 618.50 | *Thermosynechococcus elongatus* | 5B82 (100) | 26–186 | 1,196.8 | Extended pocket |
| Metals | CusS | 107.87 | *Escherichia coli* | 5KU5 (101) | 38–185 | 58.1 | |
| | CzcS | 65.39 | *Pseudomonas aeruginosa* | 5GPO (102) | 38–161 | 91.5 | |
| No cofactor or ligand binding | Agp1 (Atu1990) | NA | *Agrobacterium fabrum* | 5HSQ (103) | 20–108 | 52.7 | |
| | Agp2 (Atu2165) | NA | *Agrobacterium fabrum* | 6G1Y (104) | 21–119 | 175.1 | Pocket open to solvent |
| | AhR | NA | *Homo sapiens* | 5NJ8 (105) | 106–253 | 67.5 | |
| | AhR | NA | *Mus musculus* | 4M4X (106) | 41–186 | 42.3 | |
| | AhRR | NA | *Homo sapiens* | 5Y7Y (107) | A102–A256 | 133.0 | |
| | ARNT (PAS A) | NA | *Bos taurus* | 5Y7Y (107) | 89–189 | 1,188.7 | Open binding groove |
| | ARNT (PAS B) | NA | *Bos taurus* | 5Y7Y (107) | B208–B311 | 128.2 | |
| | ARNT (PAS B) | NA | *Homo sapiens* | 1X0O (108) | 1–119 | 38.9 | |
| | ARNT (PAS A) | NA | *Mus musculus* | 4ZP4 (109) | 92–263 | 48.1 | |
| | ARNT (PAS B) | NA | *Mus musculus* | 4ZP4 (109) | 282–384 | 137.3 | |
| | BMAL1/ARNTL (PASB) | NA | *Mus musculus* | 4F3L (110) | 277–382 | 234.3 | Pocket w/o occupancy |
| | CLOCK (PAS B) | NA | *Mus musculus* | 4F3L (110) | 250–353 | 146.5 | |
| | Cph1 | NA | *Synechocystis sp.* | 2VEA (111) | 29–126 | 65.8 | |
| | DhaR/YcgU | NA | *Escherichia coli* | 4LRX (35) | C214–C305 | 90.7 | |
| | BphP | NA | *Deinococcus radiodurans* | 1ZTU (112) | 52–144 | 79.7 | |
| | EAG/Kcnh1 | NA | *Mus musculus* | 4LLO (113) | B23–B134 | 112.5 | |
| | EAG/Kcnh1 | NA | *Rattus norvegicus* | 5K7L (114) | A27–A132 | 186.6 | |
| | PadC | NA | *Idiomarina species A28L* | 5LLW (115) | 33–123 | 86.7 | |
| | MmoS (PAS B) | NA | *Methylococcus capsulatus* | 3EWK (72) | 122–227 | 88.0 | |
| | NcoA1 PAS B | NA | *Homo sapiens* | 5NWM (116) | A254–A385 | 410.7 | |
| | NcoA-1/ SRC-1 | NA | *Mus musculus* | 1OJ5 (117) | A259–A367 | 204.5 | Pocket w/o occupancy |
| | BphP | NA | *Pseudomonas aeruginosa* | 3C2W (118) | 25–114 | 136.1 | |
| | PhyB | NA | *Arabidopsis thaliana* | 4OUR (119) | 29–131 | 41.1 | |
| | PpsR (N-PAS) | NA | *Rhodobacter sphaeroides* | 4HH2 (120) | 29–125 | 25.7 | |
| | PpsR (PAS1) | NA | *Rhodobacter sphaeroides* | 4HH2 (120) | 166–261 | 23.4 | |
| | PpsR (PAS2) | NA | *Rhodobacter sphaeroides* | 4HH2 (120) | 284–383 | 56.3 | |
| | BphP1 PAS1 | NA | *Rhodopseudomonas palustris* | 4GW9 (121) | 54–145 | 53.2 | |
| | BphP1 PAS2 | NA | *Rhodopseudomonas palustris* | 4GW9 (121) | 549–646 | 121.2 | |
| | BphP2 | NA | *Rhodopseudomonas palustris* | 4E04 (122) | 29–121 | 21.0 | |
| | BphP3 | NA | *Rhodopseudomonas palustris* | 2OOL (123) | 42–138 | 100.1 | |
| | BphP | NA | *Stigmatella aurantiaca* | 6BAF (124) | 17–112 | 12.3 | |
| | BphP2 | NA | *Stigmatella aurantiaca* | 6PTQ (125) | 19–103 | 31.4 | |

**TABLE 2** (Continued)

| Cofactor or ligand | Protein | $M_w$ of cofactor or ligand (g/mol) | Organism | PDB | PAS domain boundary from PDB RCSB | Pocket MS vol (Å³) | Comment |
|---|---|---|---|---|---|---|---|
| | Soluble guanylate cyclase (sGC) PAS α domain | NA | Manduca sexta | 4GJ4 (126) | 10–110 | 31.4 | |
| | Soluble guanylate cyclase (sGC) α subunit | NA | Homo sapiens | 6JT0 (127) | A288–A386 | 47.1 | |
| | Soluble guanylate cyclase (sGC) β subunit | NA | Homo sapiens | 6JT0 (127) | B217–B326 | 72.5 | |
| | XccBphP (N-terminal PAS domain) | NA | Xanthomonas campestris | 5AKP (128) | 33–128 | 97.4 | |
| | XccBphP (C-terminal PAS domain) | NA | Xanthomonas campestris | 5AKP (128) | 534–637 | 852.6 | Open binding groove |
| dCache - amino acids | CtaA | 89.09 | Pseudomonas fluorescens | 6PXY (129) | 41–269 | 147.3 | |
| | Mlp24/ McpX/ VC_A0923 | 89.09 | Vibrio cholerae | 3C8C (21) | 1–226 | 138.5 | |
| | Mlp37 | 105.09 | Vibrio cholerae | 5AVE (130) | 5–234 | 127.5 | |
| | PctA | 149.21 | P. aeruginosa | 5LTX (44) | 29–256 | 334.5 | |
| | PctB | 175.21 | P. aeruginosa | 5LT9 (44) | 33–256 | 227.8 | |
| | PctC | 103.12 | P. aeruginosa | 5LTV (44) | 33–257 | 191.8 | |
| | PscC | 115.13 | Pseudomonas syringae | 6MNI | 23–275 | 192.5 | |
| | Tlp3 | 131.17 | Campylobacter jejuni | 4XMR (131) | 37–285 | 248.9 | |
| | TlpQ | 111.14 | P. aeruginosa | 6FU4 (132) | 39–323 | 753.7 | Open binding groove |
| dCache - cytosine | Dret_0059 | 111.10 | Desulfohalobium retbaense | 5ERE (133) | 322–562 | 349.1 | |
| dCache - phosphate | VP0354 (vpHK1S-Z8) | 94.97 | Vibrio parahaemolyticus | 3LID (21) | 8–269 | 224.1 | |
| dCache - polyamines | McpU | 88.15 | Pseudomonas putida | 6F9G (134) | 41–300 | 615.0 | |
| Cache - no cofactor or ligand binding | LuxQ | NA | Vibrio cholerae | 3C38 (21) | 21–240 | 37.0 | |
| | LuxQ | NA | Vibrio harveyi | 2HJE (135) | 2–221 | 23.7 | |
| dCache - QACs | McpX | 144.19 | Rhizobium meliloti | 6D8V (136) | 38–306 | 229.7 | |
| dCache - cytokinins | AHK4 | 203.24 | Arabidopsis thaliana | 3T4J (137) | 126–393 | 528.2 | |
| dCache - carboxylic acids | DctB | 118.09 | Rhizobium meliloti | 3E4O (138) | 48–301 | 134.1 | |
| | DctB | 118.09 | Vibrio cholerae | 3BY9 (24) | 27–285 | 130.3 | |
| | KinD | 88.06 | Bacillus subtilis | 4JGO (139) | 6–204 | 156.2 | |
| | TlpC | 90.08 | Helicobacter pylori | 5WBF (140) | 3–261 | 263.2 | |
| sCache - acetate sensing | Adeh_3718 | 59.04 | Anaeromyxobacter dehalogenans | 4K08 (47) | 57–144 | 84.9 | |
| sCache - carboxylic acids | CitA | 189.10 | Klebsiella pneumoniae | 1P0Z (50) | 50–126 | 356.9 | Pocket open to solvent |
| | DcuS | 134.09 | Escherichia coli | 3BY8 (24) | 56–130 | 174.9 | |

**TABLE 2** (Continued)

| Cofactor or ligand | $M_w$ of cofactor or ligand (g/mol) | Protein | Organism | PDB | PAS domain boundary from PDB RCSB | Pocket MS vol (Å³) | Comment |
|---|---|---|---|---|---|---|---|
| | 88.06 | VP0183 | *Vibrio parahaemolyticus* | 4EXO (45) | 56–146 | 95.9 | |
| | 73.07 | PscD-SD | *Pseudomonas syringae* | 5G4Y (46) | 32–178 | 98.7 | |
| sCache - metals | 58.69 | PhoQ | *Escherichia coli* | 3BQ8 (59) | 41–138 | 108.5 | |
| | 40.08 | PhoQ | *Salmonella enterica* serovar Typhimurium | 1YAX (23) | 39–138 | 32.8 | |
| sCache - urea | 60.05 | TlpB | *Helicobacter pylori* | 3UB6 (45) | 70–156 | 170.9 | |

aPAS or Cache domain boundaries are indicated. The pocket or cavity volume is presented along with the molecular weight ($M_w$) of the cofactor or ligand in the pocket/cavity, where present. MS, pocket volume based on the molecular surface; QAC, quaternary ammonium compound.

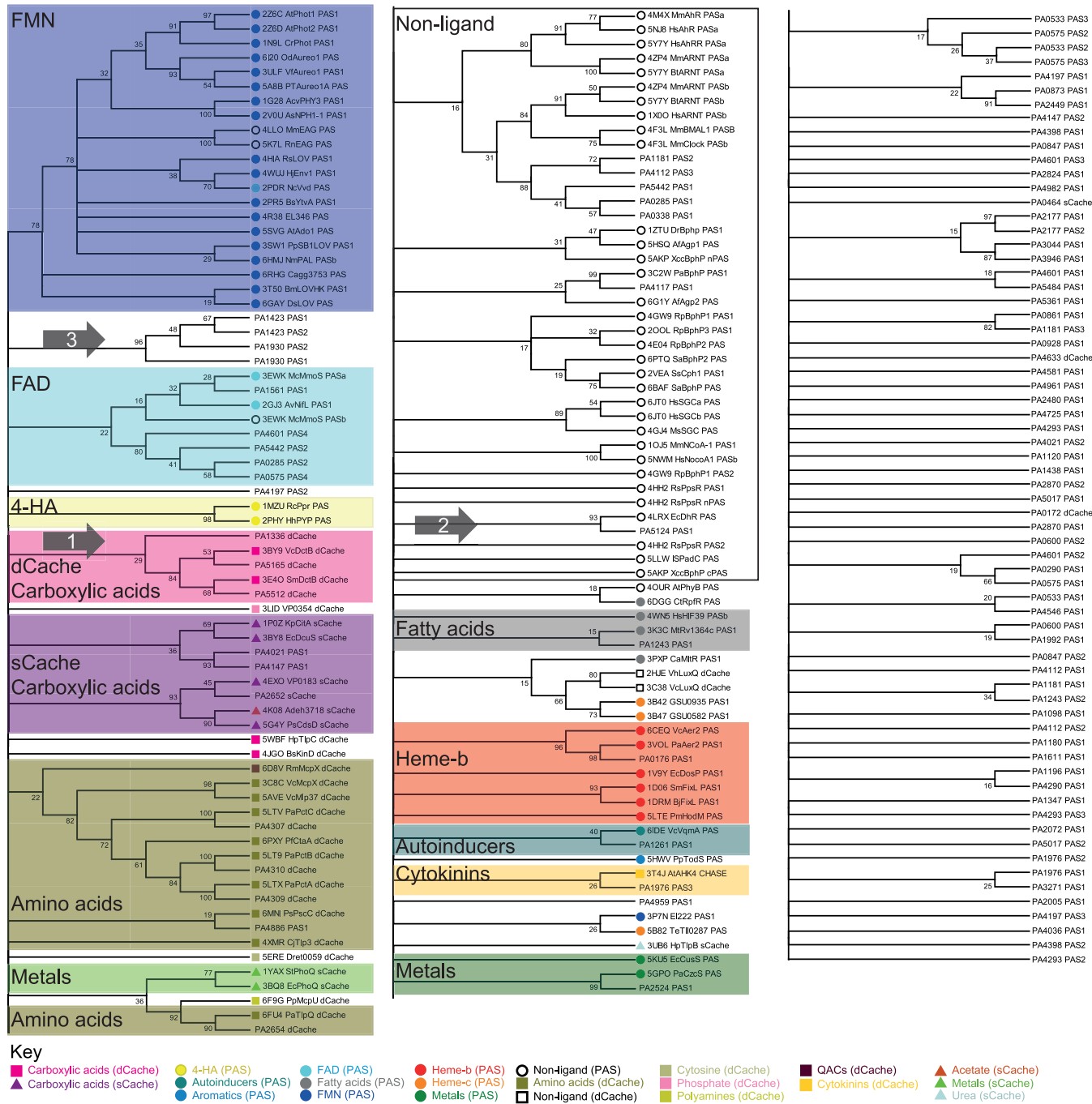

**FIG 1** Maximum likelihood phylogenetic analysis of *Pseudomonas aeruginosa* PAO1 PAS or Cache domains with the reference set of structurally characterized domains. The percentage of bootstrap replicates that reproduced each branch is given, with branches corresponding to less than 15% of bootstrap replicates collapsed and rearranged for clarity. PAS, dCache, and sCache domains are labeled with a circle, square, or triangle, respectively. The nature of ligand or cofactor is given in the key and denoted by color, and individual alignments of these groups are found in the supplemental material. Groups discussed in the text are marked with a numbered arrow. The supplement to this article contains an evaluation of different phylogenetic analyses and alignments of individual clades shown in Fig. 1 and discussed in the text.

was taken to be indicative of the capacity to bind cofactor or ligand. Cofactor or ligand binding capacity was thus inferred from phylogenetic analysis and conservation of key amino acid residues.

The dCache domains of PA1336, PA5165, and PA5512 that were identified to group with sequences of carboxylic acid-binding dCache domain structures from the reference data set (arrow 1 in Fig. 1) were investigated to confirm conservation of ligand-coordinating residues. Amino acids responsible for substrate coordination within the carboxylic acid-binding

## A

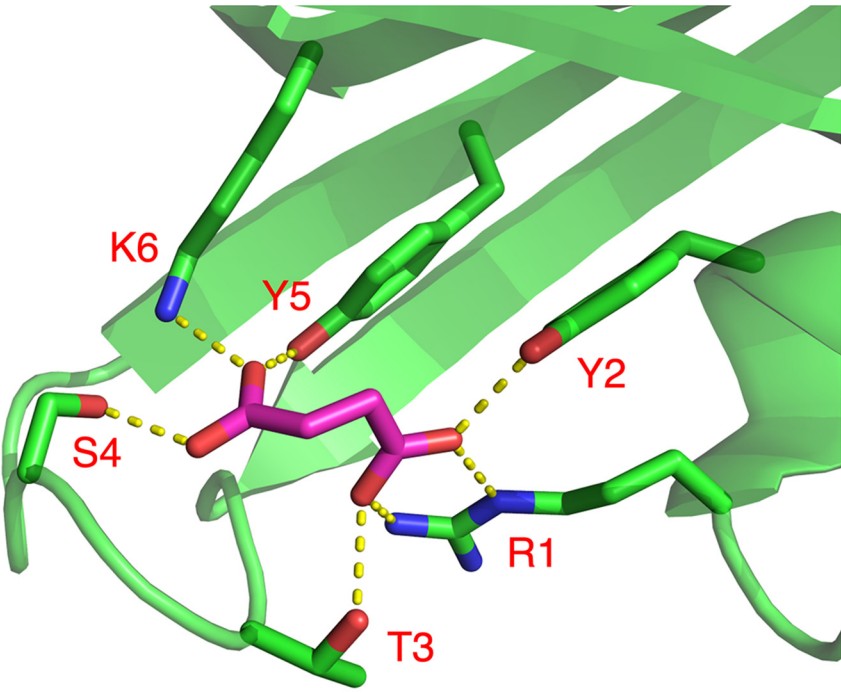

## B

FIG 2 Ligand-binding capacity of the dCache phylogenetic clade investigated by primary/secondary structure conservation analysis. Sequences selected here are highlighted by arrow 1 in Fig. 1. (A) The dCache domain of *Rhizobium meliloti* DctB in complex with succinate (PDB 3E4O). Residues involved in the coordination of succinate are shown as sticks and are labeled with single-letter amino acid codes and consecutive numbers. (B) Guided sequence alignment using the predicted secondary structure for PAO1 PA1336, PA5165, and PA5512 against the carboxylic acid-binding dCache domains from *R. meliloti* DctB, *Vibrio cholerae* DctB, and *Bacillus subtilis* KinD. The predicted secondary structure used for alignment is denoted as a cartoon under the sequences. The position of amino acids used for ligand binding in DctB is indicated in the alignment.

dCache domain of DctB are shown in Fig. 2. It is worth pointing out that these residues vary between KinD carboxylic acid-binding dCache domains, where substrates (succinate, malonate, pyruvate, and lactate) adopt a different binding pose, and DcuS sCache domains of different organisms that display different substrate specificity (citrate, malate, pyruvate, and propionate). PA5512, PA1336, and PA5165 use the same repertoire of ligand-coordinating amino acid residues as DctB. However, the slight variation needs to be discussed with respect to substrate specificity.

An interesting example evaluates fatty acid-binding PAS domains, of which a number of different binding poses and interactions have been structurally characterized (alignment found in supplemental material). The reference protein Rv1364c from *Mycobacterium tuberculosis* contains a binding motif used to coordinate palmitic acid in reference 38. PA0847 PAS2, PA1196 PAS1, PA1976 PAS3, and PA4112 PAS2 all display conservation of the two relevant amino acids responsible for side chain-specific ligand interaction. In PA1196, there is a conservative exchange (aspartate to glutamate)

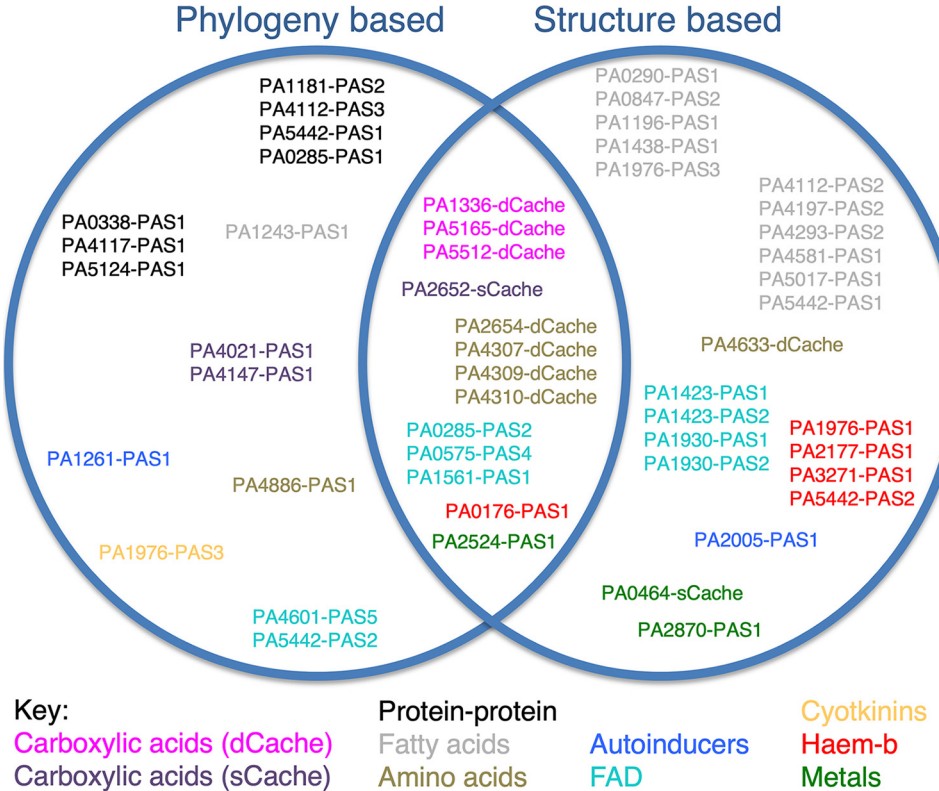

**FIG 3** Combination of phylogeny- and conservation-based assignment. *Pseudomonas aeruginosa* PAS and Cache domains predicted to bind cofactors or ligands are grouped by method of prediction. The nature of the bound cofactor or ligand is denoted by color.

in one of these two recognition amino acids. The analysis of conservation performed here adds significantly to the phylogenetic analysis, as the majority of fatty acid-binding PAS domains were assigned on the basis of conserved ligand-binding amino acids.

Indeed, a number of novel assignments can be made based on conserved binding motif. For example, PA2005 PAS1 could not be placed into a clade through phylogenetic analysis but is assigned here as autoinducer binding based on conservation of the two residues critical for side chain-specific coordination of the autoinducer DPO (3,5-dimethylpyrazin-2-ol), seen in *V. cholerae* VqmA (6IDE) (39). The alignment reveals that one amino acid is conserved while the other one is a conservative exchange from lysine to arginine. The data generated are summarized in Fig. 3 and Table 3.

## DISCUSSION

Individual domains are the building blocks of modular proteins and are required for functional diversification of the proteome. Understanding of protein function is crucially dependent on our grasp of physiological and functional roles of these constituting domains. For the omnipresent PAS and Cache domains, analysis is generally hampered by failure to predict cofactor- or ligand-binding state from sequence. Even identification of these domains proves to be difficult, due to low sequence conservation. While HMM sequence searches are a sensitive method to detect homology in cases of low sequence identity, we also made use of the SMART domain prediction server (30, 31) and used structure-guided analyses here.

We have studied the bacterial model organism *Pseudomonas aeruginosa* that has the ability to adapt to various environmental conditions, a survival strategy and an underling property important in the clinical setting. We sought to identify the nature of cofactors and ligands that bind to PAS or Cache domains within *P. aeruginosa* PAO1 using phylogeny and structural conservation analyses. A number of differences exist

**TABLE 3** PAS or Cache domains and predicted cofactors or ligands assigned on the basis of combined phylogeny and sequence-structure alignment

| Cofactor or ligand | Protein | Domain | Known physiological role |
|---|---|---|---|
| Amino acids | PA2654 (TlpQ) | dCache | Chemotaxis toward ethylene and histamine (132, 141, 142) |
| | PA4307 (PctC) | dCache | Chemotaxis toward amino acids (44, 143, 144) |
| | PA4309 (PctA) | dCache | Chemotaxis toward amino acids (44, 143–145) |
| | PA4310 (PctB) | dCache | Chemotaxis toward amino acids (44, 143, 144) |
| | PA4633 | dCache | Unknown (146) |
| | PA4886 | PAS1 | Unknown (147, 148) |
| Autoinducers | PA1261 (LhpR) | PAS1 | Transcriptional regulator (149) |
| | PA2005 (HbcR) | PAS1 | Regulation of (R)-3-hydroxybutyrate catabolism (150) |
| Carboxylic acids -dCache like | PA1336 (AauS) | dCache | Regulation of genes involved in aspartate, glutamate, and glutamine uptake and catabolism (43) |
| | PA5165 (DctB) | dCache | Regulation of $C_4$-dicarboxylic acid transport systems (34) |
| | PA5512 (MifS) | dCache | Regulation of $\alpha$-ketoglutarate transport and utilization (41, 42) |
| Carboxylic acids -sCache like | PA2652 (CtpM) | sCache | Chemotaxis toward malate (48, 49, 151) |
| | PA4021 (EatR) | PAS1 | Regulation of ethanolamine catabolism (51) |
| | PA4147 (AcoR) | PAS1 | Regulation of 2,3-butanediol and acetoin metabolism (52, 53) |
| Cytokinins | PA1976 (ErcS') | PAS3 | Regulation of ethanol oxidation (152, 153) |
| FAD | PA0285 | PAS2 | Regulation of biofilm formation (154) |
| | PA0575 | PAS4 | Regulation of biofilm formation in response to L-arginine |
| | PA1423 (BdlA) | PAS1 | Regulation of biofilm dispersal (17, 155, 156) |
| | PA1423 (BdlA) | PAS2 | Regulation of biofilm dispersal (17, 155, 156) |
| | PA1561 (Aer/TlpC) | PAS1 | Aerotaxis (157, 158) |
| | PA1930 (McpS) | PAS1 | Regulation of chemotaxis (40) |
| | PA1930 (McpS) | PAS2 | Regulation of chemotaxis (40) |
| | PA4601 (MorA) | PAS4 | Regulation of flagellar development and protease secretion (159–162) |
| | PA5442 | PAS2 | Unknown |
| Fatty acids | PA0290 | PAS1 | Regulation of biofilm formation and Psl production (154, 163–165) |
| | PA0847 | PAS2 | Regulation of motility in response to a no. of stimuli (165, 166) |
| | PA1196 (DdaR) | PAS1 | Regulation of methylarginine metabolism, role in quorum-sensing (167, 168) |
| | PA1243 | PAS1 | Regulation of swimming and biofilm formation (169) |
| | PA1438 (MmnS) | PAS1 | Regulation of efflux pump expression (170) |
| | PA1976 (ErcS') | PAS2 | Regulates ethanol oxidation (152, 153) |
| | PA4112 | PAS2 | Histidine kinase of unknown pathway |
| | PA4197 (BfiS) | PAS2 | Regulation of biofilm formation (171–174) |
| | PA4293 (PprA) | PAS2 | Regulation of outer membrane permeability/of biofilm formation (175–177) |
| | PA4581 (RtcR) | PAS1 | Homologous to *E. coli* regulator of RNA 3′-terminal phosphate cyclase expression (178–180) |
| | PA5017 (DipA) | PAS1 | Biofilm regulation, chemotaxis, motility, maintenance of c-di-GMP heterogeneity (19, 181–183) |
| | PA5442 | PAS1 | Unknown |
| Heme-b | PA0176 (Aer2/TlpG/McpB) | PAS1 | Aerotaxis and virulence (93, 184, 185) |
| | PA1976 (ErcS') | PAS1 | Regulates ethanol oxidation (152, 153) |
| | PA2177 | PAS1 | Unknown |
| | PA3271 (MxtR) | PAS1 | Redox sensing and interbacterial signaling (186, 187) |
| | PA5442 | PAS2 | Unknown |
| Metals | PA0464 (CreC) | sCache | Regulation of carbon source catabolism (188, 189) |
| | PA2524 (CzcS) | PAS1 | Regulation of metal detoxification and resistance to carbapenem antibiotics (102, 190–192) |
| | PA2870 | PAS1 | Diguanylate cyclase involved in biofilm production, Psl production, regulation of swimming motility (165) |
| No cofactor or ligand binding | PA0285 | PAS1 | Regulation of biofilm formation (154) |
| | PA0338 | PAS1 | Regulation of biofilm formation, Psl production, and swimming motility (165) |
| | PA1181 (YegE) | PAS2 | Biofilm dispersal (18, 193) |
| | PA4112 | PAS3 | Histidine kinase of unknown pathway |
| | PA4117 (BphP) | PAS1 | Quorum sensing (118, 194, 195) |
| | PA5124 (NtrB) | PAS1 | Regulation of nitrogen metabolism, rhamnolipid production, biofilm formation, expression of virulence genes, and swarming (196–200) |
| | PA5442 | PAS1 | Unknown |

between predictions based on maximum likelihood phylogeny and the individual alignment and inspection of conservation of cofactor- or ligand-interacting amino acids. The PAS domains of PA0873 and PA2449 provide an example in which phylogenetic analysis places them with the reference 4-hydroxycinnamic acid-binding PAS domains. However, when conserved ligand- or cofactor-interacting sidechains were assessed, this classification did not hold. Therefore, analysis based on one method alone may be indicative but not conclusive. The results of our combined analysis and predictions are summarized in Tables 1 and 3, and Fig. 3 highlights differences in assignment from the two different approaches used here.

Our analysis revealed a number of relationships and provides new insight. An example are the four PAS domains marked with the black arrow 3 in Fig. 1 that mark the PAS domains of PA1423 (BdlA) and PA1930 (McpS). Both proteins possess the same architecture, with two N-terminal PAS domains coupled to a methyl-accepting chemotaxis domain. Though they are clearly related, the question of functional diversification arises. Indeed, PA1930 has been reported to have a negative effect on chemotaxis (40), while PA1423 is involved in biofilm dispersal (17). It is therefore likely that the two proteins respond to different triggers and, in doing so, lead to a different biological output. We have experimentally characterized a similar example previously with the proteins PA2072 and RbdA that share an architecture but are responsible for two almost orthogonal functions (19). Thus, there are examples where gene duplication allows proteins to diversify to functionally evolve.

An interesting observation is made here with flavin binders. The distinct clade with sequences of the flavin mononucleotide (FMN)-binding PAS domain structures does not reveal PAO1 PAS domain relatives. However, FAD-binding PAS domains are identified in PAO1. Whether, indeed, FMN is not used as a cofactor in PAO1 remains to be seen. It might turn out that PAO1 has some remarkable and truly distinct PAS domains, and further structural analyses rather than predictions will in time reveal this.

The strength of the combined approach to analyze both phylogeny and conservation of cofactor- or ligand-specifying amino acids is exemplified here with the analysis of carboxylic acid-binding PAS or Cache domains. When the *in silico* results are placed into physiological context, additional insight is gained. We identified carboxylic acid-binding domains in two distinct classes for both sCache and dCache sensory architectures. Variation in substrates and their coordination is detectable between the two classes (24), and consequently, we identify different clades likely to present different substrate interaction and selectivity. The dCache domains illustrate the approach taken by combining sequence, phylogenetic, and structural information. As such, the dCache domains of PA1336, PA5165, and PA5512 are all inferred here to be able to bind carboxylic acids (Fig. 2).

The dCache domain of PA5165 (DctB) was assigned as carboxylic acid binding and, within *P. aeruginosa*, DctB acts as sensor of a two-component pathway involved in regulating the expression of $C_4$-dicarboxylic acid transport systems (34). It therefore follows that binding of carboxylic acids to the dCache domain of DctB could directly couple levels of $C_4$-dicarboxylic acids to a signaling cascade responsible for the expression of transport systems used in their uptake (Fig. 4)

Similarly, a carboxylic acid-binding dCache domain within PA5512 (MifS) could also directly link signal perception to a known phenotype (Fig. 4). MifS is required for the transport and utilization of the $C_5$-dicarboxylic acid $\alpha$-ketoglutarate (41, 42). It can therefore be hypothesized that this might reveal a potential substrate not previously recognized for dCache domains. As carboxylic acid-binding dCache domains are known to bind $C_3$ and $C_4$ substrates with at least one carboxylic acid (15), it is conceivable that the dCache domain in MifS may also be able to bind $\alpha$-ketoglutarate and act as a sensor. The variation of amino acids identified in the binding pocket from sequence alignments may reflect the required level of flexibility to accept various substrates or binding poses across the variety of PAS and Cache domains in these proteins.

The assignment of PA1336-dCache as carboxylic acid binding could help to identify a source of selectivity within dCache domains. Studies of a protein orthologous to

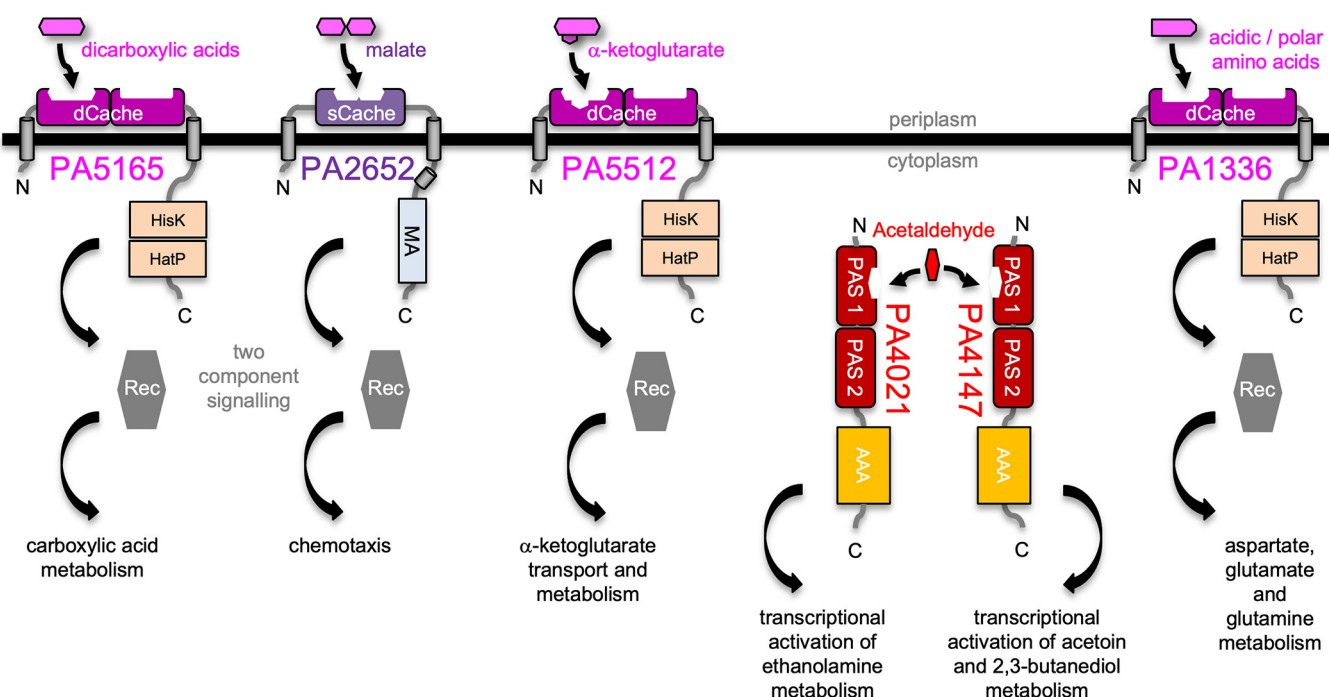

**FIG 4** Proteins assigned as containing carboxylic acid-binding Cache and PAS domains are involved in various signaling cascades. PA5165 and PA2652 bind carboxylic acids with periplasmatic dCache/sCache domains. PA5512 is involved in the transport and metabolism of α-ketoglutarate, a previously undescribed ligand for carboxylic acid-binding dCache domains. From the analysis presented here, PA1336 is predicted to bind polar and acidic amino acids. PA5165, PA5512, and PA1336 are sensor histidine kinases (HisK, His kinase A; HatP, histidine kinase-like ATPase), while PA2652 is a chemoreceptor (MA, methyl-accepting). Another cascade in response to acetaldehyde promotes transcriptional changes through interaction with cytoplasmic PAS domains in PA4021 or PA4147; this ligand has not previously been described for PAS domains.

PA1336 (AauS) within *Pseudomonas putida* demonstrate a role in utilization of the amino acids aspartate, glutamate, and glutamine (43), all of which contain side chain carboxylic acid or C=O groups. As PA1336-dCache was classified here as carboxylic acid binding and not amino acid binding, it could be speculated that a similar role for PA1336 within PAO1 to its orthologue in *P. putida* could be accommodated through interaction between the side chains of these amino acids and the PA1136-dCache domain, which then induces a conformational change that activates the two-component system partner of PA1336 to alter gene expression (Fig. 4) (43). Indeed, the highly conserved region identified for amino acid-binding dCache domains (44) is different in PA1336-dCache and instead shows similarity to carboxylic acid-binding dCache domains, with carboxyl groups likely neutralizing the charge through conservation of positively charged side chains of the amino acids labeled R1 and K6 in Fig. 2. These observations may guide future predictions of amino acid selectivity.

Another example of a straightforward link between our analysis and a previously defined physiological function would be the coupling of malate binding to PA2652 (CtpM) with chemotaxis (Fig. 4). The sCache domain of CtpM is assigned as carboxylic acid binding on the basis of its phylogenetic relationship with *Vibrio paraheamolyticus* VP0183 (4EXO) (45), *P. syringae* PscD (5G4Y) (46), and *Anaeromyxobacter dehalogenans* Adeh_3718 (4K08) (47) (Fig. 1) and conservation of five ligand-coordinating amino acids. A function in carboxylic acid binding aligns well with previous reports that CtpM is involved in chemotaxis and has substrate specificity toward malate, which is a known substrate for carboxylic acid-binding sCache domains (24, 48, 49). It may therefore follow from our analysis that the binding of malate to an sCache within CtpM directly couples malate concentration to associated chemotaxis signaling.

Interestingly, phylogeny analysis groups the first PAS domain of PA4021 (EatR) and the first PAS domain of PA4147 (AcoR) with the carboxylic acid-binding sCache domains of *E. coli* DcuS (3BY8) (24) and *Klebsiella pneumoniae* CitA (1P0Z) (50) and the

phosphate-binding dCache domain of *Vibrio parahaemolyticus* VP0354 (3LID) (21). Both PA4021 and PA4147 have known functions as transcriptional regulators for the metabolism of small, hydroxyl-containing, organic compounds and are proposed to perform those functions in response to acetaldehyde (51–53). As acetaldehyde is, to some extent, similar in structure to the carboxylic acids detected by the reference structures in these clades, it is possible that acetaldehyde binding directly to the PAS domains present in PA4021 and PA4147 could form a concise way to induce these changes in transcription and could be the basis of a novel class of PAS domain ligand (Fig. 4).

In conclusion, this study uses protein sequence comparison, phylogeny, and structure-based prediction of ligand or cofactor binding for PAO1 PAS and Cache domains. Although just predictions, the classifications presented give insight from comparison with similar proteins, leading to experimentally testable hypotheses to gain functional insights.

## MATERIALS AND METHODS

**Selection of *P. aeruginosa* PAO1 PAS and Cache domains.** HMM-to-HMM comparisons have previously identified 70 proteins within *P. aeruginosa* PAO1 that contain PAS and Cache domains (22, 26). Protein sequences of these proteins were retrieved from the *Pseudomonas* genome database (11). Selection of the final data set of 101 sequences, containing 91 PAS domains, 9 dCache domains, and 2 sCache domains, is described in Results. These are listed in Table 1.

**Generation of the reference data set with 3D structures of PAS and Cache domains.** The DALI webserver (54) was used for an exhaustive search of PAS and Cache domains within the Protein Data Bank (PDB; March 2020). Search models were chosen to represent different cofactor- or ligand-binding architectures. PAS domains from *Bradyrhizobium japonicum* FixL (heme-b binder, PDB: 1xj2 [55]), *Brucella abortus* LOV-HK (FMN binder, 3t50 [56]), *Azotobacter vinelandii* NifL (FAD binder, 2gj3 [57]), and *H. halophila* PYP (4′-hydroxycinnamic acid binder, 2phy [58]) were used. Further, sCache domains PhoQ (cation binder, 3bq8 [59]) and DcuS (carboxylic acid binder, 3by8 [24]) from *E. coli* were used. Finally, the dCache domains DctB (carboxylic acid binder, 3by9 [24]) from *V. cholerae* and PctB (amino acid binder, 5lt9 [44]) from *P. aeruginosa* were used. The structures were submitted individually and together retrieved a total of 7,513 matches, corresponding to 986 individual PDB entries. The results included structures not classified as either PAS or Cache domains that were discarded, for example, structurally related GAF domains. Retained were structures with a functional cofactor or ligand bound as well as structures with a reported signaling function, referred to hereafter as "no cofactor or ligand" binding. The final reference data set contained a total of 106 PAS and Cache domains trimmed down to the PAS and Cache domain boundaries and included 78 PAS domains, 20 dCache domains, and 8 sCache domains.

**Maximum likelihood phylogeny.** The 106 sequences of the reference data set and the 101 sequences from *P. aeruginosa* PAO1 were aligned using CLUSTALW, as implemented in MEGA7 (60, 61). This alignment was then subjected to molecular phylogenetic analysis by maximum likelihood methods within MEGA7 (61, 62). Initial phylogenetic trees were obtained by applying Neighbor-Join (63) and BioNJ (64) algorithms to a matrix of pairwise distances estimated using the JJT-matrix based model (65). The trees were scored and automatically selected based on log-likelihood scores. The bootstrap consensus tree is inferred from 100 replicates and taken to represent the evolutionary history of taxa analyzed (66).

**Sequence-structure analysis.** The *P. aeruginosa* sCache and dCache domains were aligned to the equivalent subsets in the reference data set using PROMALS3D (37) to determine conservation of ligand-coordinating amino acids residues. Conservation was used to suggest their potential ligand-binding class. For the larger set of PAS domain sequences, the reference data set was divided up according to ligand or cofactor (see Table 2) and then aligned against the *P. aeruginosa* PAS sequences.

**Determination of binding pocket or cavity size.** The sizes of enclosed cavities or of binding pockets that are open to the surrounding environment allow different classes of PAS or Cache domains to be distinguished. To map their size, the coordinates of the reference structures were uploaded to the CASTp server (67), with the PAS or Cache domain boundaries as given in Table 2. CASTp returns multiple pockets and cavities, which were inspected using UCSF Chimera (68); where a cofactor or ligand was present, this pocket was chosen, but when no cofactor or ligand was identified, the one closest to the center of the PAS or Cache domain was reported, ensuring all cavities/pockets reported here were in a similar position. Table 2 reports the solvent-excluded volume calculated with a probe sphere radius of 1.4 Å, based on Connolly's molecular surface calculation (69), as this parameter was able to discriminate ligand- or cofactor-binding pockets/cavities most clearly. Where the open pockets reported volumes that include not only the actual ligand or cofactor cavity but also the access to the cavity, this is noted in Table 2.

## SUPPLEMENTAL MATERIAL

Supplemental material is available online only.

**SUPPLEMENTAL FILE 1**, PDF file, 7.4 MB.

## ACKNOWLEDGMENTS

We acknowledge funding by Diamond Light Source and the University of Southampton to A.H. and C.C. Conceptualization, I.T., M.A.W.; investigation, A.H., C.C.; formal analysis, data curation, and validation, A.H., C.C., I.T.; visualization, A.H., I.T.; supervision, M.A.W., J.S.W., I.T.; writing – review and editing, A.H., I.T.; writing – original draft, A.H.; funding acquisition, M.A.W., J.S.W., I.T.

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
