## [Reviewer comments · Microbiology Spectrum]

Microbiology Spectrum

Phylogenetic analysis with prediction of cofactor or ligand binding for *Pseudomonas aeruginosa* PAS and Cache domains

Ivo Tews, Charlotte Cordery, Martin Walsh, Jeremy Webb, and Andrew Hutchin

Corresponding Author(s): Ivo Tews, Biological Sciences, Institute for Life Sciences (IfLS)

Review Timeline:

Submission Date:	July 26, 2021
Editorial Decision:	September 1, 2021
Revision Received:	October 29, 2021
Accepted:	November 21, 2021

Editor: Gaurav Sharma

Reviewer(s): Disclosure of reviewer identity is with reference to reviewer comments included in decision letter(s). The following individuals involved in review of your submission have agreed to reveal their identity: Michael Y Galperin (Reviewer #1)

Transaction Report:

DOI: <https://doi.org/10.1128/spectrum.01026-21>

September 1, 2021

Dr. Ivo Tews
Biological Sciences, Institute for Life Sciences (IfLS)
University Southampton
Southampton SO17 1BJ
United Kingdom

Re: Spectrum01026-21 (**Phylogenetic analysis with prediction of cofactor binding for *Pseudomonas aeruginosa* PAS domains**)

Dear Dr. Ivo Tews,

Thank you for submitting your manuscript to Microbiology Spectrum. This is a well-executed project that provides several useful insights for the community. Two reviewers have gone through this manuscript and they have recommended reconsideration of your paper following major revision. Based on their opinions, I invite you to resubmit your manuscript after addressing all reviewer comments. The comments of the reviewers are included at the bottom of this letter.

The first major comment from Reviewer 2 is the most critical one, therefore, please give more emphasis on this comment. Altogether, please address all the comments by both reviewers. Also, pay particular attention to the comments about clear terminology. Please ensure that the added comments are well explained throughout the text and supported by clearly described methods.

When submitting the revised version of your paper, please provide (1) point-by-point responses to the issues raised by the reviewers as file type "Response to Reviewers," not in your cover letter, and (2) a PDF file that indicates the changes from the original submission (by highlighting or underlining the changes) as file type "Marked Up Manuscript - For Review Only". Please use this link to submit your revised manuscript - we strongly recommend that you submit your paper within the next 60 days or reach out to me. Detailed information on submitting your revised paper is also provided below.

Link Not Available

Sincerely,

Gaurav Sharma, Ph.D.
Editor, Microbiology Spectrum

Journals Department
Reviewer comments:

Reviewer #1 (Comments for the Author):

The manuscript by A. Hutchin and colleagues provides a sequence- and structure-based classification of PAS and Cache domains encoded in signaling proteins from *Pseudomonas aeruginosa* strain PAO1. This classification allows the authors to predict the range of ligands that could be bound - and sensed - by each respective protein.

The manuscript is well-drafted and describes the methods used and conclusions drawn in sufficient detail. I have only minor comments that the authors might want to address in revision.

I would also encourage the authors to make sequence alignments of the eight colored clusters in Fig. 1, either as supplementary material to this paper.

Minor comments

L. 59. Change 'make' to 'makes'

L. 60. Change 'leading' to 'causing'

L. 61. Remove 'Therefore'

L. 74-75. While the N-terminal helix does cross the membrane, it is most likely the C-terminal helix of this domain that propagates the signal.

L. 96. "they are more sensitive" More sensitive than what?

L.172. "The similar architecture facilitates a discussion" Where is this discussion?

Fig. 4. In the legend, change "The four cascades involve two-component signaling" to "PA5165, PA5512, and PA1336 are sensor histidine kinases, while PA2652 is a chemoreceptor". I am afraid that the use of the same shape for the histidine kinase (HisK/HATPase), methyl-accepting (MA-sensor), and receiver (REC) domains is somewhat misleading.

Reviewer #2 (Comments for the Author):

In this paper, Hutchin et al. used protein sequence and phylogenetic analyses to classify and predict ligand/cofactor binding for *Pseudomonas aeruginosa* PAS and Cache domains. Predictions derived by the authors can be tested by experimentalists working with this model organism. The authors collected and summarized a comprehensive body of information related to ligand and cofactor binding by PAS and Cache domains. This is a very positive aspect of the paper. Computational predictions are technically sound, but without experimental validation of any of them, they remain just "predictions".

Major concerns:

1. As evident from the title and throughout the manuscript, the authors consider Cache domains as a version of PAS domains. This is incorrect and Ref. 20 provides a clear distinction between these two domain superfamilies. Therefore, "PAS and Cache domains" should be used instead of "PAS domains" in the title and throughout the manuscript, as appropriate (e.g. PAS and Cache, not PAS/CACHE).
2. In a nearly similar fashion, "cofactor" and "ligand" are not the same thing. The authors should clearly specify that. For example, in the title and in the heading on line 174, it should be stated "... cofactor and ligand binding..."
3. The authors generated an unusually large bibliography list, and, in many instances, they give full credit to both original discoveries and reviews. Surprisingly, they failed to reference the original discovery of the PAS domain superfamily (by two independent groups, see PMID: 9301332 and 9382818).
4. Tree building methods other than maximum likelihood (e.g., neighbor-joining, Bayesian) should be used for comparison.

Minor concerns:

1. CACHE should not be capitalized, it is "Cache".

Staff Comments:

Preparing Revision Guidelines

Please return the manuscript within 60 days; if you cannot complete the modification within this time period, please contact me. If

you do not wish to modify the manuscript and prefer to submit it to another journal, please notify me of your decision immediately so that the manuscript may be formally withdrawn from consideration by Microbiology Spectrum.

If you would like to submit an image for consideration as the Featured Image for an issue, please contact Spectrum staff.

Response to Reviewers

Manuscript Spectrum01026-21

REVIEWER REPORT(S): EDITOR

Thank you for submitting your manuscript to Microbiology Spectrum. This is a well-executed project that provides several useful insights for the community. Two reviewers have gone through this manuscript and they have recommended reconsideration of your paper following major revision. Based on their opinions, I invite you to resubmit your manuscript after addressing all reviewer comments. The comments of the reviewers are included at the bottom of this letter.

The first major comment from Reviewer 2 is the most critical one, therefore, please give more emphasis on this comment. Altogether, please address all the comments by both reviewers. Also, pay particular attention to the comments about clear terminology. Please ensure that the added comments are well explained throughout the text and supported by clearly described methods.

Done. In response to comment 1 from reviewer 2, the nomenclature has been updated throughout the manuscript to clarify the difference between PAS and Cache domains.

When submitting the revised version of your paper, please provide (1) point-by-point responses to the issues raised by the reviewers as file type "Response to Reviewers," not in your cover letter, and (2) a PDF file that indicates the changes from the original submission (by highlighting or underlining the changes) as file type "Marked Up Manuscript - For Review Only". Please use this link to submit your revised manuscript - we strongly recommend that you submit your paper within the next 60 days or reach out to me. Detailed information on submitting your revised paper is also provided below.

REVIEWER REPORT(S): Referee: 1

The manuscript by A. Hutchin and colleagues provides a sequence- and structure-based classification of PAS and Cache domains encoded in signaling proteins from Pseudomonas aeruginosa strain PAO1. This classification allows the authors to predict the range of ligands that could be bound - and sensed - by each respective protein.

The manuscript is well-drafted and describes the methods used and conclusions drawn in sufficient detail. I have only minor comments that the authors might want to address in revision.

Thank you.

I would also encourage the authors to make sequence alignments of the eight colored clusters in Fig. 1, either as supplementary material to this paper.

Done. We have included secondary structure annotated alignments of clades highlighted in Figure 1 and discussed in the text in a new supplement. We reference this in the main text several times and introduce the alignments presented in supplement at the end of the phylogeny section, as follows:

“Alignments of individual clades are presented in Supplement, and we give a few examples in the following section.”

Minor comments

L. 59. Change 'make' to 'makes'

Done.

L. 60. Change 'leading' to 'causing'

Done.

L. 61. Remove 'Therefore'

Done.

L. 74-75. While the N-terminal helix does cross the membrane, it is most likely the C-terminal helix of this domain that propagates the signal.

Thank you very much, corrected to:

“Typically acting as signal receptors, they bind small ligands and propagate signals into the cell interior, suggested to be mediated by the C-terminal helix that crosses the membrane (19-22).”

L. 96. "they are more sensitive" More sensitive than what?

Thank you, updated and changed to:

“Hidden Markov Models (HMM) were used as a sensitive method for homology detection, and these methods are typically employed for cases with low sequence identity (26).”

We also updated the same point in the discussion (L. 237), which now reads:

“While HMM sequence searches are a sensitive method to detect homology in cases of low sequence identity, we also made use of the SMART domain prediction server (28, 29) and used structure guided analyses here.”

L.172. "The similar architecture facilitates a discussion" Where is this discussion?

Thank you very much, updated to read:

“The similar architecture suggests functional differentiation for these proteins.”

Fig. 4. In the legend, change "The four cascades involve two-component signaling" to "PA5165, PA5512, and PA1336 are sensor histidine kinases, while PA2652 is a chemoreceptor".

Done.

I am afraid that the use of the same shape for the histidine kinase (HisK/HATPase), methyl-accepting (MA-sensor), and receiver (REC) domains is somewhat misleading.

Done, updated.

REVIEWER REPORT(S): Referee: 2

In this paper, Hutchin et al. used protein sequence and phylogenetic analyses to classify and predict ligand/cofactor binding for Pseudomonas aeruginosa PAS and Cache domains. Predictions derived by the authors can be tested by experimentalists working with this model organism. The authors collected and summarized a comprehensive body of information related to ligand and cofactor binding by PAS and Cache domains. This is a very positive aspect of the paper. Computational predictions are technically sound, but without experimental validation of any of them, they remain just "predictions".

Thank you. We agree with the referee that predictions based on sequence analysis require experimental validation and have spelled this out in the conclusion sentence of the revised discussion:

“In conclusion, this study uses protein sequence comparison, phylogeny, and structure-based prediction of ligand/cofactor binding for PAO1 PAS/CACHE domains. Although just “predictions”, the classification presented gives insight from comparison with similar proteins, leading to experimentally testable hypotheses to gain functional insights.”

Major concerns:

1. As evident from the title and throughout the manuscript, the authors consider Cache domains as a version of PAS domains. This is incorrect and Ref. 20 provides a clear distinction between these two domain superfamilies. Therefore, "PAS and Cache domains" should be used instead of "PAS domains" in the title and throughout the manuscript, as appropriate (e.g. PAS and Cache, not PAS/CACHE).

Thank you, updated throughout.

2. In a nearly similar fashion, "cofactor" and "ligand" are not the same thing. The authors should clearly specify that. For example, in the title and in the heading on line 174, it should be stated "... cofactor and ligand binding..."

Thank you, updated throughout.

3. The authors generated an unusually large bibliography list, and, in many instances, they give full credit to both original discoveries and reviews. Surprisingly, they failed to reference the original discovery of the PAS domain superfamily (by two independent groups, see PMID: 9301332 and 9382818).

We believe that including reviews along with original citations is justified to direct the reader not only to the original sources but also to inferences made through comparison.

We thank the referee for the suggestion to include the two papers that are milestones in establishing the PAS domain clade.

4. Tree building methods other than maximum likelihood (e.g., neighbor-joining, Bayesian) should be used for comparison.

Done. We have included this comparison in a new supplement for further reading:

“To establish the phylogeny of PAS and Cache domains we show a maximum likelihood tree in this paper. We sought to compare this tree to phylogenetic trees constructed using different models, namely neighbor-joining; maximum parsimony with a Subtree-Pruning-Regrafting algorithm; minimum evolution with an initial tree generated using a neighbor-joining algorithm, and searched using the Close-Neighbor-Interchange algorithm; and an unweighted pair group method with arithmetic mean (UPGMA). In all cases sequences aligned using CLUSTALW, as implemented in MEGA7, identical to the analysis presented in the main paper, before subsequent tree generation using MEGA7. The reader may evaluate these trees for differences ...”

The supplement delivers an analysis that compares different ways of constructing phylogenetic trees. This analysis is referenced from the main text in the third paragraph of the phylogenetic results chapter, as follows:

“The maximum likelihood phylogeny analysis with a 100-replicate bootstrap consensus tree is shown in Figure 1. For the PAS and Cache domains analyzed here, we found maximum-likelihood grouped PAS or Cache domains from the reference dataset into clades of similar cofactor or ligand binding across the largest number of bootstrap replicates, in comparison to other ways of constructing phylogenetic trees (see SI for further detail).”

Minor concerns:

1. CACHE should not be capitalized, it is "Cache".

Done, though CACHE it is typically capitalized in the literature.

November 21, 2021

Dr. Ivo Tews
Biological Sciences, Institute for Life Sciences (IfLS)
University Southampton
Southampton SO17 1BJ
United Kingdom

Re: Spectrum01026-21R1 (**Phylogenetic analysis with prediction of cofactor or ligand binding for *Pseudomonas aeruginosa* PAS and Cache domains**)

Dear Dr. Ivo Tews,

It is a pleasure to accept your manuscript entitled "Phylogenetic analysis with prediction of cofactor or ligand binding for *Pseudomonas aeruginosa* PAS and Cache domains" in its current form for publication in Microbiology Spectrum. With this email, I am forwarding it to the ASM Journals Department for publication. You will be notified when your proofs are ready to be viewed.

Sincerely,

Gaurav Sharma
Editor, Microbiology Spectrum
<https://sites.google.com/view/sharmaglab/>
